# Microevolution and Its Impact on Hypervirulence, Antimicrobial Resistance, and Vaccine Escape in *Neisseria meningitidis*

**DOI:** 10.3390/microorganisms11123005

**Published:** 2023-12-18

**Authors:** August Mikucki, Charlene M. Kahler

**Affiliations:** 1Marshall Centre for Infectious Diseases Research and Training, School of Biomedical Sciences, University of Western Australia, Perth, WA 6009, Australia; august.mikucki@telethonkids.org.au; 2Wesfarmers Centre of Vaccines and Infectious Diseases, Telethon Kids Institute, University of Western Australia, Perth, WA 6009, Australia

**Keywords:** *Neisseria meningitidis*, genetic diversity, microevolution, vaccine escape

## Abstract

*Neisseria meningitidis* is commensal of the human pharynx and occasionally invades the host, causing the life-threatening illness invasive meningococcal disease. The meningococcus is a highly diverse and adaptable organism thanks to natural competence, a propensity for recombination, and a highly repetitive genome. These mechanisms together result in a high level of antigenic variation to invade diverse human hosts and evade their innate and adaptive immune responses. This review explores the ways in which this diversity contributes to the evolutionary history and population structure of the meningococcus, with a particular focus on microevolution. It examines studies on meningococcal microevolution in the context of within-host evolution and persistent carriage; microevolution in the context of meningococcal outbreaks and epidemics; and the potential of microevolution to contribute to antimicrobial resistance and vaccine escape. A persistent theme is the idea that the process of microevolution contributes to the development of new hyperinvasive meningococcal variants. As such, microevolution in this species has significant potential to drive future public health threats in the form of hypervirulent, antibiotic-resistant, vaccine-escape variants. The implications of this on current vaccination strategies are explored.

## 1. The Pathogenic *Neisseria* and Their Relationship to the Human Host

The genus *Neisseria* contains ten described species known to colonise the urogenital, oral, dental, and pharyngeal surfaces of humans [1]. Eight of these species, *N. lactamica*, *N. bacilliformis*, *N. mucosa*, *N. cinerea*, *N. elongata*, *N. oralis*, *N. polysaccharea*, and *N. subflava*, are exclusively commensals and only cause disease in immunocompromised patients. The remaining two species, *Neisseria gonorrhoeae* (Ngo) and *Neisseria meningitidis* (Nme), are able to cause disease in humans. Ngo is considered an obligate pathogen and causes the sexually transmitted infection gonorrhoea, which presents as urethritis with urethral discharge and dysuria in men [2] and is commonly asymptomatic in women [3]. Long-term gonococcal infection can result in invasive disease including epididymo-orchitis in men, pelvic inflammatory disease in women, or disseminated gonococcal infection in both sexes [4]. Infection with Ngo results in an influx of neutrophils to the site of infection (Figure 1A). Since the gonococcus is naturally able to resist killing by neutrophils and multiply inside the phagosome, translocation of infected neutrophils back to the mucosal surface serves as a vector for the transmission of gonorrhoea [5]. Nme, on the other hand, is a transient commensal of the nasopharynx (Figure 1B) and is only rarely pathogenic, causing invasive meningococcal disease (IMD) [6].

Nme normally colonises the nasopharyngeal epithelium in humans and is transmitted by aerosolised droplets or contact with infected fluids such as saliva. IMD occurs when the meningococcus crosses the nasopharyngeal epithelium and enters the bloodstream, resulting in the rapid onset of septicaemia and/or meningitis [7]. Since infection of neither the blood nor meninges improves the transmission of the meningococcus, IMD is considered to be an evolutionary dead end for this organism [8]. The events which initiate the development of IMD from carriage are not fully understood but include host factors such as viral infection and environmental factors such as climate [9]. The meningococcal population itself is not uniform in its ability to cause disease, with some isolates being associated entirely with carriage and others being associated more commonly with disease. The ability of a given meningococcal lineage to cause disease can be given by the odds ratio of the association of that lineage with disease over carriage (defined as the disease/carriage ratio, D/C ratio) [10]. Large studies assessing meningococcal carriage and disease across multiple countries have found a broad diversity in D/C ratio among meningococcal genetic lineages, with some being exclusively associated with carriage and others frequently causing disease [11,12]. An understanding of how meningococcal virulence evolved and how virulence is related to the genetic diversity of this organism has been a key area of study.

## 2. Meningococcal Diversity, Epidemiology, and Invasive Potential

The earliest typing scheme developed for the meningococcus was based on agglutination reactions against the polysaccharide capsule using immune rabbit serum. On this basis, thirteen serogroups were defined, and twelve of these (serogroups A, B, C, E, H, I, K, L, W, X, Y, and Z) were later confirmed and given defined structures using nuclear magnetic resonance spectroscopy (Figure 2) [13]. More recently, whole-genome sequencing allowed the assignment of Nme into genogroups based on the genetic organisation of the *cps* island, which encodes for capsule synthesis, with each genogroup corresponding to the presumed capsular serogroup. The *cps* island is organised into six regions: region A, which encodes genes for capsular polysaccharide synthesis; region B, which encodes the capsule translocation system; region C, which is divergently transcribed from region A and encodes for capsule transport proteins; region D, which contains the *galE* locus important for LOS synthesis; region D’, which is a duplicate copy of region D with a truncated *galE2* locus; and region E, which contains the gene of unknown function *tex* and two methyltransferase pseudogenes [13,14]. Nonserogroupable meningococci do occur and belong to one of two broad groups: those that possess a *cps* island that has been rendered nonfunctional due to indels or IS element insertion in key capsule synthesis genes or, more commonly, those which lack a *cps* island entirely. The latter group instead possesses a *cnl* locus consisting of regions D and E only. Since a capsule is required to cause disease in immunocompetent individuals [15], serogrouping remains an important tool for tracking meningococcal epidemiology.

Capsule switching is the process by which a meningococcal isolate of a given serogroup acquires the ability to express an alternative capsular polysaccharide [16]. To do this, meningococci must acquire an alternative set of genes within region A of the *cps* island which encodes the genes for synthesis of capsule monosaccharide components and the capsule polymerase [16]. Capsule switching is mediated by recombination, and the *cps* island may be an example of a minimal mobile element which was acquired fairly recently in the ancestral meningococcal population by HGT from the family *Pasteurellaceae* or a similar ancestor [17,18]. Analysis of the *cps* island from a number of meningococcal isolates suggests that regions D, D′, and E are hotspots for recombination, and studies determining the recombination breakpoints of specific capsule-switching events frequently observe recombination between regions D and A or regions D and B in order to meditate switching of the gene content of region A [19,20]. Capsule switching occurs frequently in the meningococcal population, especially between serogroups with similar genetic organisation [21,22]. For example, C↔B and W↔Y capsule-switching events require the exchange of the capsule polymerase gene only, since these serogroups all have an otherwise similar region A encoding for the synthesis of sialic acid (Neu5Ac) which makes up all (MenB and MenC) or part (MenW and MenY) of their capsule polysaccharide [23]. Capsule-switching events involving larger regions of the *cps* island can result in capsule switching between more dissimilar serogroups. Capsule switches have been frequently reported within cc11 (C↔B↔W↔Y), cc5 (A↔X↔C), cc2057 (B↔Y), and cc4821 (C↔B↔W) [19,20,24,25,26,27,28,29,30,31,32,33,34,35].

In addition to serogrouping, multilocus enzyme electrophoresis (MLEE) typing was developed to group meningococci into enzyme types (ET). This technique depends on the gel mobility of large soluble enzymes encoded by the meningococcus, which changes based on the amino acid sequence, and therefore allele, of each enzyme [36,37]. The combination of alleles detected for a given strain can be used to assign an ET, with an ET correlating with the evolution of these large enzymes and therefore genetic relatedness of the organism as a whole. In 1998, MLEE was superseded by multilocus sequence typing (MLST), which was originally developed for the meningococcus but is now in use for a large number of bacterial pathogens [38]. MLST works on the same principle as MLEE, with alleles assigned based on the nucleic acid sequences of genes rather than inferred from gel mobility. The meningococcal MLST scheme uses the alleles of internal fragments of seven housekeeping genes (*abcZ*, *adk*, *aroE*, *gdh*, *pdhC*, and *pgm*) to assign a unique sequence type (ST) to meningococcal strains. STs which share four or more common alleles are grouped into a single clonal complex named after the putative ‘ancestral’ ST [39]. The general acceptance of the MLST scheme among researchers has enabled the development of a central database (https://pubmlst.org/organisms/neisseria-spp) containing thousands of Nme isolate records, which is an invaluable tool for tracking the epidemiology and population genetics of the meningococcus. The advent of readily available whole-genome sequencing has accelerated study of meningococcal genomics even further, since it removes the need for PCR amplification and sanger sequencing to generate an MLST type, and entire genomes can be made available online for study.

Despite the identification of thousands of meningococcal sequence types belonging to dozens of clonal complexes, isolates retrieved from IMD cases are dominated by a small subset of lineages. Almost all disease is caused by Nme from one of six serogroups (MenA, MenB, MenC, MenW, MenX, and MenY) and belonging to one of eleven clonal complexes (cc1, cc4/5, cc8, cc11, cc18, cc32, cc41/44, cc103, cc269, cc334, and cc461). These clonal complexes are termed the hyperinvasive (also hypervirulent) lineages. The hyperinvasive lineages are associated most often with disease and have a D/C ratio > 1, while carriage lineages such as cc53 are associated exclusively with carriage and have D/C ratios close to 0. Other lineages, such as cc22, cc23, cc60, cc162, cc174, cc213, cc364, and cc4821, are associated less strongly with disease and have ratios between zero and one [10,40].

Globally, the incidence of IMD follows an epidemic pattern with the repeated emergence, persistence, and eventual replacement of dominant hyperinvasive clonal complexes within a geographic region. For example, MenA isolates from cc5 were responsible for waves of epidemics in the 1990s and 2000s in Africa [41,42]. More recently, MenW isolates from cc11 (MenW:cc11) were responsible for a series of global epidemics between 2000 and 2020 after their initial emergence in South America [43,44]. The diversity of the meningococcus displays a strong geographic association, shaped by bottlenecks which occur with the introduction of an epidemic genetic lineage into a region [42]. In Europe, the majority of IMD cases are caused by MenB, with MenW, MenC, and MenY also causing disease [45]. MenC and MenW previously accounted for a greater share of disease in the region, but incidence declined following successful vaccination campaigns using the conjugate polysaccharide vaccine against MenACWY [46,47,48]. North America exhibits similar serogroup prevalence to that of Europe, with a slightly higher proportion of MenC cases in the United States [45]. In South America, MenB and MenW are by far the most common causes of IMD, in part due to the high prevalence of MenW:cc11 isolates [45,49]. Less routine surveillance data are available for countries in Asia. In China, MenW, MenC, MenA, and MenX have been reported [27,50,51], and in Japan, MenY, MenC, and MenB dominate [52]. In both Australia and New Zealand, MenB has been the predominant cause of disease, with a recent spike in MenW incidence due to the global expansion of MenW:cc11 [45]. The area of the world with the highest rates of IMD is a group of 24 countries spanning sub-Saharan Africa known as the ‘African meningitis belt’, which experience periodic epidemics where the incidence of IMD can exceed 100 cases per 100,000 population [41]. IMD in the African meningitis belt was historically predominated by MenA, but the success of the MenA vaccine in the region has seen the replacement of MenA with MenC, MenW, and MenX [45].

## 3. Mechanisms Producing Meningococcal Variation

As a human-restricted organism, the meningococcus spends its entire existence under intensive selective pressure to colonise the host epithelium, persist, and disseminate to other hosts. During this cycle, the meningococcus needs to compete against the established microflora of the host and defend itself against the innate and adaptive immune responses, both of which can vary widely from person to person [6]. Meningococcal fitness therefore relies on the ability to adapt rapidly to its environment. Despite this requirement, Nme possesses a relatively small genome and encodes only 36 predicted transcriptional regulators and four two-component systems [53]. By contrast, *Escherichia coli* encodes over 200 regulatory proteins. To compensate for a restricted regulatory framework, meningococci (and other *Neisseria* spp.) have a range of mechanisms which allow them to vary their gene expression and antigenic structure (Figure 3).

### 3.1. Natural Competence and Recombination

Like many bacterial species, *Neisseria* species are naturally competent for the uptake of foreign DNA, allowing them to incorporate these sequences into the genome by homologous recombination (Figure 3A). However, unlike most naturally competent species, *Neisseria* spp. express a high level of constitutive natural competence [54]. Neisserial competence is dependent on the presence of a specific DNA uptake sequence (DUS) being present on the incoming DNA. This allows *Neisseria* spp. to selectively uptake DNA from the same species or closely related species. DNA uptake is primarily mediated by the minor pilin and competence protein ComP, which is able to mediate DUS-dependent and DUS-independent DNA uptake, although the latter occurs much less efficiently than the former [55]. The pathogenic *Neisseria* have a DUS sequence of 5′-ATGCCGTCTGAA-3′, which is shared by *N. lactamica*, *N. polysaccharea*, and *N. cinerea* [56]. Uptake occurs most efficiently when incoming DNA contains the DUS from the same species; however, uptake may still be mediated by similar DUS from related species. The DUS is present at high frequencies in Nme genomes (>1200 copies), and the frequency of DUSs in a given region of the genome is associated with the frequency of recombination at that locus [56,57]. DUSs are significantly associated with the core genome, and especially with genes encoding DNA repair, recombination, restriction modification systems (RMS), and DNA replication [57,58]. This association, along with the fact that competence is induced by DNA damage in other bacterial pathogens, led to the hypothesis that competence in these species is largely a mechanism for the uptake of highly homologous DNA in order to mediate DNA repair in regions encoding functions essential to bacterial survival [59]. Natural competence and recombination also provide a source of alternative alleles, genomic islands, and other variable DNA which the meningococcus can use to vary its accessory genome and antigenic repertoire. Significant sequence diversity is observed for the *fetA*, *porA*, *fHbp*, *nadA*, and *nhba* loci, all of which encode adhesins or major outer membrane proteins and some of which are vaccine antigens [60,61,62,63,64,65,66,67,68,69,70,71]. Sequence variation in these genes and the spread of novel variants through recombination between lineages may allow the meningococcus to evade host immunity induced either by natural exposure or vaccination. Similarly, many of the hypervariable loci in the meningococcal genome discussed below are reliant on recombination to function.

In addition to DUS copies, the *Neisseria* genome contains a number of other repeat families which form regions around the genome with high levels of similarity to each other. Since recombination is highly dependent on homology, these regions form recombination hotspots and in some cases mediate genetic mobilisation. The most common of these after the DUS are dRS3 repeats, which are a group of 20 bp repeats with a conserved 6 bp inverted repeat [58]. There are nearly 700 dRS3 copies in the meningococcal genome [72]. Importantly, dRS3 repeats serve as insertion sites for the integration of two filamentous phages: the neisserial filamentous phage (Nf1) and the meningococcal-disease-associated (MDA) phage [73,74]. The MDA phage in particular is associated with hypervirulent clonal complexes, including cc11, and has been shown to increase the invasive potential of these isolates [75]. dRS3 elements are found flanking a family of RS elements which are between 70 and 200 bp in length [72]. Together, these elements make up the neisserial intergenic mosaic element (NIME) of which there are between 1 and 60 copies of 117 different families. NIMEs are found at greater frequencies in association with genes encoding cell-surface-associated proteins such as the transferrin and lactoferrin uptake systems TbpAB and LbpAB, the pilus adhesins PilC1 and PilC2, and the outer membrane porin PorA [76]. As such, these repeat arrays are believed to promote recombination in regions encoding surface antigens to mediate escape of host immunity.

In addition to smaller repeat arrays, larger repeat units known as Correia elements (CE) are also found throughout the meningococcal genome [77,78]. CEs are mobilisable and there are approximately 150 CE copies in the meningococcal genome [79]. CEs may encode promoter elements and are involved in the differential expression of both the LOS sialyltransferase Lst and the MtrCDE efflux pump between Nme and Ngo [80,81]. CEs are associated with a number of loci encoding opacity proteins in Ngo and have also been associated with the disruption of RNA-mediated regulatory networks [79,82]. Another family of mobilisable elements in neisserial genomes are insertion sequences (IS), which are mobile elements encoding their own transposase [58]. The majority of IS elements in Nme are from the IS*5*, IS*NCY*, and IS*30* families. A member of the IS*30* family, IS*1655*, is found only in meningococcal genomes and is thought to be associated with the speciation of Nme [18]. IS elements are often associated with genomic islands in Nme and may either promote recombination in their vicinity or act in pairs to form large transposable elements which may transfer intervening genomic material between neisserial genomes [78].

### 3.2. Phase Variation

Phase variation is the process by which bacteria may stochastically switch expression of a given gene or phenotype on or off. Phase variation may progress by several mechanisms including reversible chromosomal rearrangements, but in *Neisseria*, the most common mechanism is that of slipped-strand mispairing (Figure 3B) [83]. Slipped-strand mispairing occurs during DNA replication at tracts of repeated nucleotides termed simple sequence repeats (SSRs) [84]. During replication, the DNA polymerase may ‘slip’ when complementary base pairing occurs within SSRs between the template strand and the newly synthesised strand, resulting in the deletion or insertion of repeat units [85,86]. Variation in the length of SSRs can impact gene expression in two major ways. If the SSR is within the coding sequence of the gene, changes to the number of repeats may result in a frameshift mutation and lead to premature termination of translation and therefore the production of a nonfunctional protein [83]. This form of phase variation results in switching between normal levels of gene expression (on) and the complete absence of expression (off). The second mechanism occurs when the SSR is within the promoter region of the gene. In this instance, changes to the length of the SSR may result in the lengthening or shortening of the distance between regulatory elements in the gene’s promoter. Such changes can impact the efficiency of interactions between the promoter region and the RNA polymerase or transcription factors, thereby modulating gene expression [83]. This form of phase variation results in changes to the level of gene expression but is usually not responsible for complete loss of function. Phase variation due to slipped-strand mispairing is rapid and results in the meningococcal population being heterogeneous for a diverse array of phenotypes. As a result, the population always contains members who are ‘pre-adapted’ to circumstances requiring different levels of gene expression which can then go on to survive genetic bottlenecks.

One of the most striking features of neisserial genomes is the frequency of SSRs within both the coding sequences of genes and their promoter regions. Neisserial SSRs range from mononucleotide repeats up to 15 bases in length to extremely long tetrameric and pentameric repeats consisting of over 20 repeat units [87,88]. SSRs with repeating units consisting of up to nine nucleotides exist but are rarer. Estimates of the repertoire of phase-variable genes in Nme have found over 110 phase-variable genes [89,90]. In a study by Siena, D’Aurizio, Riley, Tettelin, Guidotti, Torricelli, Moxon, and Medini [89], stochastic variation in the length of SSRs associated with 115 meningococcal genes was readily observable in single colonies grown overnight. Phase variation is classically known to be common in genes encoding the expression of immune-visible, surface-exposed structures [91]. As such, multiple meningococcal genes encoding Lipid A and lipooligosaccharide (LOS) biosynthesis (*lgtA*, *lgtC*, *lpxB*), peptidoglycan biosynthesis (*murF*, *murG*), pilin adhesins (*pilC1*, *pilC2*), pilin glycosylation *(pglA*, *pglE*, *pglH*, *pglG*), iron acquisition systems (*tbpB*, *lbpB*, *hpuA*, *hmbR*), porins (*porA*), and the opacity proteins (*opaA*, *opaB*, *opaD*, *opaJ*) are subject to phase variation [83,91]. The genes for capsule polysaccharide synthesis in some serogroup B and C isolates (*cssA*) are phase-variable, as are the capsule *O*-acetylation genes encoded by serogroup W and Y meningococci [92]. Since capsule expression can inhibit the interaction of meningococcal adhesins with host cells, phase variation represents one mechanism by which such inhibition may be relieved. In addition to surface structures, phase variation has been associated with genes of diverse function including nucleic acid biosynthesis, DNA metabolism and repair, protein folding, and transcription factors [91].

An additional mechanism through which phase variation can modulate gene expression is through changes in DNA methylation. Three type III restriction modification systems (RMSs) with phase-variable methyltransferase genes (*modA*, *modB*, and *modD*) were identified in the pathogenic *Neisseria* [93]. Epigenetic regulation of gene expression by methylation of the promoter region is well established in a number of bacterial species [94,95,96]. In Nme, phase variation of *modA11* produced altered expression of 80 genes including 5 encoding surface-exposed proteins [97]. The cohort of genes under transcriptional control of a phase-variable methylase are known as the phasevarion. In many cases, the gene encoding the restriction endonuclease associated with phase-variable Mods is subject to frameshift mutations or deletions that result in loss of function [98]. Presumably this allows rapid switching of Mod expression without the risk of autorestriction. The presence of phasevarions in *Neisseria* allows rapid switching between two regulatory states, and since some strains carry multiple Mod systems, several combinatorial states are theoretically available to the organism to respond to multiple environmental situations [98]. Since the combination of Mod systems and their associated phasevarions vary between strains, it is likely that Mods contribute to the diversity of regulatory networks observed in the meningococcal population.

### 3.3. Antigenic Variation and Hypervariable Loci

In addition to simply switching gene expression on or off or modulating expression by phase variation, meningococci have a range of mechanisms to vary the structure of their immune-visible surface antigens. This tends to result in a set of loci, often called hypervariable loci, in the meningococcal genome being hotspots of diversity (Figure 3C) [6]. Diversity at hypervariable loci is thought to provide a diverse repertoire of epitopes for meningococcal surface antigens, and switching between them allows the evasion of the adaptive immune response raised against specific variants.

The type IV pilus (tfp) of Nme is a major exposed-surface antigen and is the first point of contact between Nme and host cells [99]. The pilus is composed of repeating protein subunits which are assembled in the periplasm and exported through the pilus machinery in the outer membrane by a complex consisting of over 20 different proteins [100]. The pilus itself is composed primarily of the major pilin PilE, with the incorporation of the minor pilins PilX, PilV, and ComP and the PilC adhesin [101]. Two classes of PilE have been observed. Class I pili are expressed by all Ngo isolates and some Nme lineages (including cc32, cc41/44, cc213, cc269, cc461, and a number of carriage-associated lineages), with the remainder of Nme lineages expressing class II (including cc5, cc11, cc22, cc23, and cc60) [102]. Class I pili are hypervariable and undergo rapid gene conversion through homologous recombination between the *pilE* locus and an array of silent *pilS* pseudogenes present downstream of *pilE* [103]. Gene conversion between *pilE* and *pilS* requires the presence of a guanine quadruplex structure encoded by a G4 motif in the promoter of *pilE* as well as expression of a small RNA (sRNA) [104]. Transcription of *pilE* initiates within the G4 motif, and the newly synthesised sRNA is able to form an R-loop by base-paring with the C-rich DNA strand [104,105]. This terminates transcription and potentiates recombination into *pilE* through an as-yet-unknown mechanism. By contrast, class II pili have far less variable PilE sequences [102]. Instead, class II pilin is characterised by an increased density of glycosylation sites. Since many of the pilin glycosyltransferase genes are phase-variable (including *pglA*, *pglE*, *pglH*, and *pglG*), meningococci expressing class II pilin can modulate their antigenic expression on pilin through variation in the structure of the pilin glycan [102]. Arrays of silent cassettes similar to *pilS* are also present in the loci encoding the Tps and Maf toxin–antitoxin systems [106]. In these systems, these silent cassettes encode alternative C-terminal domains for the toxin proteins encoded by *tpsA* and *mafB*, enabling the modulation of the target specificity of these proteins [107,108].

Following pilus-mediated adhesion, the major surface proteins involved in the intimate binding of *Neisseria* to host cells are the opacity proteins Opa and Opc [109]. Opc is invariant but is absent from some meningococcal lineages including the hypervirulent cc11. By contrast, Opa proteins are highly variable. Nme typically have four *opa* loci (*opaA* (NEIS1719)*, opaB* (NEIS1551), *opaD* (NEIS1403), and *opaJ* (NEIS0903)), while Ngo strains have eleven [72,110]. Opa proteins have eight transmembrane domains and four surface-exposed loops. Two of these loops (termed HV1 and HV2) are hypervariable, and a third (SV) is semi-variable. Frequent intragenomic recombination occurs between the *opa* loci of a single meningococcal strain, which promotes the generation of a diverse array of combinations of SV, HV1, and HV2 [111]. Meningococci can also acquire new *opa* variants through horizontal gene transfer from other Nme strains or potentially other *Neisseria* spp. [111]. In addition to enabling escape against host anti-Opa antibodies, the amino acid sequence of HV1 and HV2 are known to determine Opa receptor specificity [112,113,114]. Maintaining a diverse repertoire of Opa sequences may therefore potentiate infection in a wider range of cell types as is observed in Ngo [115]. Since Opa loci are also phase-variable due to the presence of a CTTCC repeat in the 5′ end of the gene, expression of each Opa protein encoded by each locus may be independently phased on or off. Meningococci frequently express a single Opa, which may help avoid antibody response against multiple Opa proteins or avoid the fitness cost of expressing two Opa proteins at once [116].

### 3.4. Hypermutation

Many of the described mechanisms of variation, including homologous recombination and slipped-strand mispairing, are counteracted by the mismatch repair system. Two components of this repair system, MutS and MutL, are critical for its anti-recombination activity in bacteria. MutS detects DNA mismatches and MutL has endonuclease activity [117]. Loss-of-function mutations in MutS or MutL result in a loss of mismatch repair function and a corresponding increase in the mutation rate of the bacterial cell. Mutants lacking MutSL are known as hypermutators (Figure 3D) and have been identified in multiple bacterial pathogens including *Pseudomonas aeruginosa* [118], *E. coli* [119], *Burkholderia pseudomallei* [120], and *Acinetobacter baumannii* [121]. Hypermutators play a role in the evolution of bacterial pathogens by rapidly generating diversity through their increased rates of recombination and mutation, allowing the pathogen to more rapidly adapt to challenges such as herd immunity or antibiotic selection [122]. Hypermutators have been observed in both Ngo and Nme and have been associated with outbreaks of IMD [123,124]. Mutation of *mutSL* is associated with increased phase variation [125], increased pilin antigenic variation [126], and increased variation of surface antigens [127].

## 4. Meningococcal Population Structure

Bacterial populations exhibit a wide variety of structures from essentially monomorphic species (e.g., *Salmonella enterica* s. Typhi, *Mycobacterium tuberculosis*, *Bacillus anthracis*, and *Yersinia pestis*), in which the population is essentially a single variant, to highly panmictic populations (e.g., *N. gonorrhoeae*, *Helicobacter pylori*), in which horizontal gene transfer is so frequent as to completely obscure phylogenetic signals of clonal descent [128]. In between these two extremes are a number of populations with varying degrees of clonality, in which significant levels of recombination occur but in which genetic and evolutionary barriers drive the maintenance of clones [129,130]. This is the case for a number of human pathogens including *Streptococcus pneumoniae*, *E. coli*, *Haemophilus influenzae*, and Nme [131].

The population structure of Nme has been a matter of debate for some time. The meningococcal population is highly clonal, characterised by the persistence of dominant clonal complexes over decades (Figure 4). Despite this, frequent recombination is observed both between meningococcal lineages and with other *Neisseria* spp., which should theoretically obscure the clonal signal and produce a panmictic population structure like that observed in Ngo [132,133]. One early explanation proposed for this apparent contradiction was that Nme displays an ‘epidemic’ population structure, in which the population is essentially panmictic but is subject to repeated bottlenecks which restrict the population and give the appearance of clonality [134]. However, this model cannot explain the persistence of the hypervirulent lineages over the course of almost a century of observation, nor can it explain the deeply clonal signature recovered from whole-genome phylogenetic analyses of the species [135]. A study by Jolley et al. [136] found support for a model in which new variants with a transmission advantage spread quickly but are eventually curtailed by the development of herd immunity. Another study by Buckee et al. [137] similarly found that under a model incorporating interstrain competition and immune selection, hyperinvasive lineages arose and persisted in association with small changes to transmission efficiency. This model posits that the hyperinvasive lineages persist due to their combination of high-fitness genes/alleles and generate a ‘cloud’ of less fit variants through recombination and hypervariability [137]. This cloud of variants presents an opportunity for the generation of a new combination of alleles which may produce a new high-fitness variant and go on to expand. Since recombination in Nme is so frequent, the clonal population structure is maintained primarily by selection in this model.

An alternative explanation for meningococcal clonality is that there are restrictions in gene flow between lineages which prevent the loss of clonal signal normally associated with high rates of recombination. Tibayrenc and Ayala [135] argued for a ‘predominantly clonal evolution’ model with strongly restrained recombination. Consistent with this hypothesis is the observation that meningococcal lineages differ greatly in their repertoire of RMSs [57]. Since RMSs recognise and restrict differently methylated DNA, differences in these repertoires would result in less frequent recombination between lineages than within lineages and the maintenance of clonal structure [138,139]. Budroni, Siena, Hotopp, Seib, Serruto, Nofroni, Comanducci, Riley, Daugherty, Angiuoli, Covacci, Pizza, Rappuoli, Moxon, Tettelin, and Medini [57] found that a core-genome phylogeny of major hyperinvasive lineages was associated with differences in RMS loci as well as differences in genes encoding the Tfp, DNA metabolism, toxin–antitoxin systems, two-partner secretion systems, adhesins, and cell-envelope metabolism. In their analysis, several clonal complexes fell into single phylogenetic clades, including cc11/cc8 and cc32/cc269. An analysis using a greater number of meningococcal genomes by Mullally, Mikucki, Wise, and Kahler [40] similarly observed that meningococcal lineages differed greatly in their repertoire of genomic islands and that the number of genomic islands correlated with the D/C ratio. Their study replicated the relationships observed by Budroni, Siena, Hotopp, Seib, Serruto, Nofroni, Comanducci, Riley, Daugherty, Angiuoli, Covacci, Pizza, Rappuoli, Moxon, Tettelin, and Medini [57] and observed on a larger scale that the disease-associated clonal complexes were grouped into two genogroups, each with their own distinct signature of genomic islands and a common set of genomic islands which distinguish them from carriage-associated lineages such as cc53. Similarly, an analysis by Schoen, Kischkies, Elias, and Ampattu [8] identified distinct metabolic signatures for carriage and disease-associated meningococcal lineages, with lactate metabolism, oxidative stress response, the denitrification pathway, and glutathione metabolism being linked to pathogenicity. This is consistent with the predictions of the predominantly clonal model in which the meningococcal population displays clonal associations at every level forming nested groups of ‘near-clades’ [135].

## 5. Microevolution in Nme

Discussions of the evolution of the meningococcal population structure form the basis for understanding why some meningococcal lineages are associated with IMD or carriage. Early attempts to identify a single factor or group of factors which lead to pathogenicity in the meningococcus were unsuccessful since all of the major virulence factors apart from the capsule were either found to be shared by carriage-associated Nme lineages or commensal *Neisseria* spp. or not universally present in the hyperinvasive meningococcal lineages. Capsule expression itself is necessary but not sufficient to cause IMD, since many carriage-associated Nme lineages express a capsule. Instead, meningococcal virulence appears to be the product of adaptations for increased transmission fitness in a subset of Nme lineages which have invasive potential. As previously described, the combination of metabolic and virulence alleles in a strain is crucial for its fitness. Thus, a study of the mechanisms by which meningococci undergo minute changes in their genetic makeup during colonisation, infection, and transmission during epidemics has the potential to deepen our understanding of meningococcal pathogenicity. The following sections describe studies to date which have looked at meningococcal microevolution (summarised in Figure 5).

### 5.1. Within-Host Evolution

Within-host evolution (WHE) refers to the process in which a bacterial pathogen adapts to colonisation in a given host without regard for the overall transmission fitness of the organism within the host population as a whole (Figure 5A) [140,141]. Under the WHE hypothesis, virulence evolves coincidentally through selection for the colonisation of unoccupied niches such as the intracellular space or the bloodstream [142]. In Nme, WHE is postulated to occur during the initial phases of colonisation and select for variants which successfully evade host immunity and colonise the epithelium but would not be predicted to select for variants with increased fitness in the invasive context.

Several studies have studied WHE of the meningococcus during the transition from carriage to disease state. In a case of accidental laboratory infection with a hypermutator isolate, comparative whole-genome sequencing of the laboratory isolate with the isolate retrieved from blood culture revealed a duplication of the MDA phage, loss of the pilin glycosyltransferase PglA due to phase variation, phase variation of the iron acquisition genes *hmbR* and *hpuA*, and deletion of the *lgtB* gene resulting in the loss of the terminal galactose residue on LOS [143]. Multiple variants of PilE and Opa were present in the blood culture isolate, as would be expected due to the hypervariability of these antigens. A transcriptomic analysis of the two strains showed changes in the expression of multiple genes including *nalP* (encoding an autotransporter), *pilT* (encoding a pilus retraction ATPase), two ribosomal components, and several other metabolic genes. In another study comparing paired isolates from throat swabs and blood culture in four IMD patients, phase variation of *pilC1* and *pglI* and gene conversion events in *pilE* were identified [91]. Since the majority of nonsilent mutations affected the Tfp, throat and blood isolates differed slightly in their adhesion to Detroit 562 epithelial cells but did not differ in their resistance to human serum. Since the Tfp is essential for both colonisation of the nasopharynx and attachment and manipulation of the microvasculature endothelium [144], the authors suggested that strains adapted to the epithelial niche may be preadapted for invasivity. In a large study comparing paired blood and cerebrospinal fluid isolates of Nme and *S. pneumoniae*, no evidence for adaptation between these two compartments was observed [145]. Together, these studies indicate that the major changes observed in invasive isolates are those associated with the Tfp and other adhesins. This supports the hypothesis that WHE in the meningococcus is primarily a byproduct of adaptation for colonisation rather than selection for a virulent phenotype as such.

In addition to comparisons between carriage and disease isolates, some studies compared meningococcal isolates during persistent carriage in the same host (>1 month). A study by Alamro et al. [146] examined the phase variation states of major outer membrane proteins during persistent carriage in 21 individuals. They observed phase variation in *fetA*, *nadA*, *opc*, *porA*, *hpuA*, and *nalP* but noted that expression of the iron acquisition protein HmbR and adhesin/autotransporter MspA did not change during the experiment. Of particular interest was the observation that carriage strains decreased their expression of phase-variable genes over time, particularly *fetA* and *nadA*. This suggests that higher expression of these antigens may be selected for during colonisation with an eventual switch to selection for lower expression as the host mounts an immune response against them. This idea is reinforced by measurements of the anti-PorA and anticapsule antibody titres in carriers, which display an increase during acquisition and remain high during persistence [146]. A similar study by Green et al. [147] identified that horizontal gene transfer was relatively more frequent than de novo mutation in a group of 25 carriers. They also identified directional allelic variation in *pilE* as well as directional changes in expression between Opa proteins and from PilC1 to PilC2 due to phase variation. Since variation in expression of variant Opa and PilC proteins impacts receptor specificity, the persistent changes at these loci may be indicative of a change in tissue tropism during persistent carriage [147]. In a large study comparing persistent carriage, transmission, and invasive disease isolates from US high school students, paired carriage isolates from the same individual differed by a mean of 10.7 loci [148]. There was a strong correlation in this study between genes which evolved during persistent carriage, those that evolved during transmission, and those that evolved during the transition to invasive disease. In keeping with other studies, the genes which changed most frequently were *pilE*, the *opa* loci, and the *modA12* methyltransferase gene. Genes involved in surface structures (*pglEAHI*, *tspA*, *cpsJ,* and *nalP*), the *modB* methyltransferase gene, and iron regulation genes (*hpuA*, *frpC*) were also variable. A subset of genes underwent microevolution frequently during transmission but achieved fixation during persistent carriage including pilin biogenesis and glycosylation genes (*pilHTUQ*, *pglD*), the LOS sialyltransferase gene *lst*, additional surface proteins (*apbC* and *pbp2*), iron uptake genes (*tbpA* and *lbpA*), the gene encoding the IgA1 protease, and metabolic genes (*adhA*, *clpA*, *icd*, *pabB*). A third set of genes encoding metabolic functions and homologous recombination (*fumC*, *adh2*, *hsdR*, and *priA*) were more variable during the transition from carriage to disease. A study of persistent carriage in Ethiopia similarly identified *pilE*, *opa* loci, pilin glycosylation genes, *hpuA*, and *tspA* as the most variable genes during carriage [149]. A similar pattern of genetic changes was observed in a controlled infection study using *N. lactamica* [150]. In this study *pilE*, pilin glycosyltransferase genes, *hpuA*, *fetA*, and a type I RMS were the most variable during carriage over a one-month period. Taken together, these studies again reinforce that microevolution during meningococcal carriage is largely concentrated in surface structures known to be involved in colonisation and entry into host epithelial cells, as well as those proteins which are immune-visible. The impact that these genetic changes have on the host–pathogen interactions of the meningococcus in a given host is a subject requiring further study.

### 5.2. Microevolution during Epidemics and Outbreaks

In addition to changes associated with WHE, microevolution of meningococcal lineages can be observed at a population level (Figure 5B). A number of studies have examined genetic changes during meningococcal outbreaks or epidemics caused by hyperinvasive meningococci.

Early studies identified variation in the alleles of surface antigens within hyperinvasive meningococcal lineages [42,151,152,153,154]. Both reassortment of *opa* alleles within a lineage and acquisition of new HV1 and HV2 variants from outside of the lineage is observed within a relatively short timeframe. One study also identified the acquisition of Opa variants typically associated with Ngo by epidemic MenA isolates [151]. The majority of variants in these studies did not persist, appearing only once, whereas others would reach fixation and become widespread within months or years [42]. One study which tracked successive epidemics caused by MenA observed nine ‘genoclouds’, each made up of a commonly isolated genotype and a group of highly similar genotypes which were more rarely observed [153]. The study also found an association between the emergence of new outbreak genotypes and the acquisition of new *opa*, *tbpB*, and *pgm* alleles as well as IS element insertion. The authors suggest that this pattern may reflect the emergence of new high-fitness variants due to reassortment within the existing lineage, and that the ‘cloud’ of associated genotypes may represent escape variants which have a temporary advantage in hosts with immunity to the dominant genotype [153]. In a study examining evolution within the cc11 lineage (at the time referred to as the ET-37 complex), chromosomal rearrangements were identified between ET-37 and ET-15 isolates (identified by Lucidarme, Hill, Bratcher, Gray, du Plessis, Tsang, Vazquez, Taha, Ceyhan, Efron, Gorla, Findlow, Jolley, Maiden, and Borrow [44] as sublineages 11.1 and 11.2, respectively) [155]. Studies of microevolution in ET-15 identified allelic changes in *porA* and the acquisition of alternate *pilA* and *pilD* alleles from endemic meningococcal lineages during the progression of outbreaks in the Czech Republic and Australia [155,156].

More recent studies have been able to perform in-depth analysis of genomic changes thanks to the advancement of whole-genome sequencing technologies. Studies looking at isolate-level diversity within outbreaks routinely identify pilin antigenic variation, modulation of surface antigens including PorA and FetA, and changes to the pilin glycosylation gene repertoire [157,158,159,160]. As in earlier studies, population bottlenecks and strong geographical signals are commonly observed. In one study, endemic MenA ST-2859 isolates (a member of cc5) were completely replaced by MenW ST-2881 isolates before reemerging as a highly clonal outbreak [161]. The authors suggest that competition with introduced MenW isolates caused a genetic bottleneck and that the reemerged ST-2859 clone may represent selection for a high-fitness variant of this lineage. The acquisition of genetic material by outbreak strains is also commonly reported, with donors ranging from other hyperinvasive meningococcal lineages to carriage lineages and other *Neisseria* spp. [158,159,162]. A detailed study of the emergence of MenW ST-2881 isolates in the African meningitis belt identified 17 recombination blocks [159]. These recombination events involved 61 genes encoding a number of surface-associated and metabolic genes including *tbpAB*, *lgtAE*, *lst*, *pilC1*, *pilPOMN*, *ftsY*, and *dnaJ*. Of particular note were two recombination events affecting *yggS-pilT-pilU*, which regulates expression of Tfp. Interestingly, during the short timeframe (approximately one year), there was no variation observed in the *porA*, *porB*, *fetA*, *nhba*, and *opcA* loci, which are normally hypervariable. This may be indicative of selection for this combination of alleles in this genetic background and host population. A third theme of outbreak-evolution studies is the development of distinct clades within a clonal complex or ST. Studies examining MenA:cc5, MenW ST-2881, and MenB:cc32 isolates all identified the emergence of persistent subclades which underwent clonal expansion following recombination events [157,158,159,162].

A lineage in which evolution is particularly well studied is cc11, which is a hypervirulent lineage associated historically with outbreaks of IMD caused primarily by MenC and occasionally by MenB meningococci. The cc11 lineage is of particular interest due to the emergence of MenW:cc11 isolates, which have become a major cause of IMD on a global scale since the beginning of the 21st century [163]. MenW:cc11 isolates are thought to have arisen due to a capsule-switching event from a MenC:cc11 ancestor in the later decades of the 20th century [19]. The cc11 lineage is divided into two sublineages: lineage 11.1, which contains ‘classical’ ET-37 MenC:cc11 isolates and the vast majority of MenW:cc11 isolates, and lineage 11.2, which contains ET-15 MenC and MenB isolates possessing a characteristic *fumC* polymorphism and the IS*1301* element [164]. MenW:cc11.2 isolates are further divided into a number of strains which have been associated with epidemics and outbreaks: the Hajj strain, which caused the first major MenW:cc11 outbreak in 2000; the Endemic South African strain; the Burkina Faso/North African Strain; and the South American/UK strain. The South American/UK strain is responsible for outbreaks in multiple countries across all inhabited continents. Analysis of recombination within the MenW:cc11.2 lineage identified changes in the alleles of *nadA, nhbA, porB, aniA, norB,* and *fHbp* which were the result of recombination events within the lineage and were variably present in each cluster [19]. The success and high invasive potential of the MenW:cc11.1 lineage may in part be due to changes in these critical virulence genes including *fHbp*, which is important for serum resistance, and *aniA* and *norB*, which play a role in the oxidative stress response and may facilitate nitrate-dependent anaerobic growth. It is interesting to note that cc11.2 isolates commonly have an internal stop codon in *aniA* and *fHbp* and an insertion sequence disrupting *nadA*. The changes at these loci in cc11.2 may reflect differences in the selective pressures acting on these two lineages. Another noteworthy feature of cc11 is that it is commonly associated with unusual IMD presentations including pneumonia, myocarditis, and urethritis [165,166,167,168]. Isolates from cc11.2 were found to be the cause of an outbreak of urethritis among men who have sex with men in Europe and North America [169,170]. The lineage responsible for this outbreak was associated with a gonococcal-like phenotype including loss of capsule expression and restoration of *aniA* and *fHbp* expression [171]. This clade was found to have acquired gonococcal genomic material in three recombination events including one affecting *aniA* and *norB* [171]. These genomic changes are thought to have allowed this meningococcal clade to adapt to the urogenital tract by enabling growth under anaerobic conditions. Collectively, the natural history of cc11 suggests that nitrate metabolism and the oxidative stress response, as well as complement resistance, may be particularly important for the niche adaptation and virulence of this lineage.

### 5.3. Microevolution in Response to Antimicrobials

Beta-lactam antibiotics (including penicillin, ampicillin, and ceftriaxone), fluoroquinolones (such as ciprofloxacin), and chloramphenicol have all been used as empirical treatment for IMD, with ceftriaxone being the usual antibiotic of choice [172,173]. Unlike Ngo, in which multidrug-resistant strains are common, relatively low levels of resistance are observed in Nme overall despite recent reports of beta-lactamase-expressing strains [173,174]. One possible explanation for the different rates of resistance observed in these closely related organisms is their difference in lifestyles. Since Ngo is obligately pathogenic, significant selective pressure is placed on the organism during antibiotic treatment of gonorrhoea. By contrast, disease symptoms are not a necessary part of the transmission cycle for the meningococcus, and so treatment of IMD will generally not result in the dissemination of resistant variants into the population. Under these conditions, Nme is usually only exposed to antimicrobials during treatment of other infections in a colonised person, resulting in slower evolution of resistance traits.

The primary class of antibiotics for which resistance has been recorded in Nme are the beta-lactams. Although resistance through beta-lactamase expression was recently reported [175], the most common mechanism is provided by mutations in penicillin-binding protein 2 (PBP2), the primary target of these antibiotics [176]. Mutations in the *penA* gene encoding PBP2 result in the production of a protein with reduced affinity to penicillin or ceftriaxone, thus conferring resistance [176]. Rather than arising through single-point mutations, resistant PBP2 alleles are thought to be produced by recombination within the *penA* locus, producing mosaic alleles with a number of polymorphisms which each together confer resistance [177]. These alleles have been identified to have homology with *penA* alleles found in commensal *Neisseria* spp. and are hypothesized to have been transferred to the pathogenic *Neisseria* though recombination [178].

Resistance to non-beta-lactam antibiotics used to treat IMD is rare in the meningococcal population. However, it is worth noting that the extensive array of resistance mechanisms observed in the gonococcus all have the potential to be conferred to Nme strains by recombination given the high level of genetic similarity between these organisms [179,180]. The majority of these resistance mechanisms arise through point mutation or acquisition of resistance genes [174]. However, several are noteworthy examples of microevolution. Resistance to azithromycin in Ngo may be conferred by the A2059G and C2611T mutations in the 23S ribosomal RNA (rRNA), which reduce the binding affinity of azithromycin and other macrolides to the bacterial ribosome [181,182]. The gonococcus has four rRNA copies throughout the genome, and mutation at only a single locus is insufficient to produce high-level resistance. However, in the presence of selection using macrolide antibiotics, these mutations are transferred to all four rRNA loci and convey high-level resistance [182]. Overexpression of the MtrCDE efflux pump produces resistance to several antimicrobials including azithromycin and extended-spectrum cephalosporins by increasing their export from the cell [174,183]. Overexpression can be achieved by several mechanisms including mutations or deletion of the MtrR repressor, changes to the *mtrR* or *mtrCDE* promoter regions, or mosaicism in the *mtrRCDE* region through a similar mechanism to that described for mosaic PBP2s [174].

### 5.4. Microevolution in Response to Vaccination

Vaccination is a primary means of prevention and control for IMD. There are two types of meningococcal vaccines commonly in use: conjugate polysaccharide vaccines, of which a quadrivalent vaccine covering MenACWY (brand names Nimenrix^®^, Menactra^®^, and Menveo^®^) is most commonly used, and outer-membrane-vesicle-based multivalent protein vaccines (brand names Bexsero^®^ and Trumenba^®^), which are used primarily to control MenB disease since the MenB capsule is an autoantigen [184].

Like antibiotics, vaccines impose a selective pressure on bacteria. This is particularly true when a species is host-restricted and when vaccines provide protection against colonisation, as is the case for Nme. Although the past two decades have been an enormous success story for the control of IMD by vaccination, it remains to be seen how the meningococcal population will change in response to this increased immune pressure. Vaccine escape is thought to be more difficult for pathogens to evolve than resistance to antimicrobials for several reasons. Firstly, the prophylactic nature of vaccination tends to prevent the establishment of large populations within the host, and smaller populations are less likely to evolve resistance. Secondly, while antimicrobial therapies tend to target pathogens through specific and targeted mechanisms, vaccines induce an immune response from the host which may enact killing of the pathogen along multiple pathways [185,186]. This is especially true for vaccines containing multiple antigens, which will allow the production of immune responses against multiple epitopes on the same pathogen.

Despite these barriers, concerns have been raised about vaccine escape in a number of bacterial pathogens of both humans and animals (Table 1). Serotype replacement is well established following the use of conjugate polysaccharide vaccines against *Streptococcus pneumoniae* [187,188,189]. In addition, capsule switching of vaccine serotypes to nonvaccine types (NVTs) in the context of vaccination has been observed on multiple occasions [190,191]. Similarly, vaccination against *Haemophilus influenzae* serotype b has been followed by an increase in the incidence of disease caused by serotype a and serotype b isolates which have lost the ability to express their capsule [192,193,194,195]. Vaccine escape has recently been observed in *Bordetella pertussis* and *Corynebacterium diphtheriae*, both of which have otherwise been successfully controlled by vaccination for decades [196]. Vaccine escape in *B. pertussis* is thought to be due to a rise in strains lacking the immunodominant pertactin antigen and has become such a concern that a pertactin-deficient vaccine is already under investigation [197,198]. A recent increase in diphtheria cases has been observed globally, especially in India, and while incomplete vaccine uptake is thought to be the major driver, significant variation in the toxin against which the vaccine is directed may play a role in the generation of vaccine-escape variants in the near future [199]. In aquaculture, vaccine-escape variants of the fish pathogens *Yersinia ruckeri* and *Streptococcus iniae* are routinely observed due to loss of flagella expression and capsule antigenic variation, respectively [200,201,202,203]. Phase variation of the glycosyltransferase genes which determine the lipopolysaccharide structure in *Pasteurella multocida* has also been associated with vaccine escape and subsequent outbreaks of fowl cholera in layer hen populations [204]. A striking commonality between all these species is a highly diverse, often clonal population and a propensity for natural transformation (as in *S. pneumoniae* and *H. influenzae*). By contrast, the vast majority of vaccines which have not seen concerns of vaccine escape to date are those used situationally in the context of outbreaks (such as the cholera vaccine) or in high-risk or postexposure settings (Q-fever vaccine, tetanus vaccine, etc.).

Concerns about vaccine escape have also been raised in the context of meningococcal vaccinations. Capsule switching is of primary concern for the conjugate polysaccharide vaccines, whereas antigen variation and/or loss of expression of the major components may lead to escape from MenB vaccination. Of particular concern is the potential for the emergence of a hypervirulent MenB lineage (for example, cc41/44, cc32, or a capsule-switched cc11) which could potentially undergo antigenic shift to escape coverage by the MenB vaccines [164]. Despite these concerns, relatively few studies have inspected the genomic changes which occur in the meningococcal population under high levels of vaccine selective pressure.

One consideration for the potential of meningococcal vaccine escape is that vaccines must place a selective pressure on carried meningococci since isolates causing IMD do not take part in transmission. Prevention of meningococcal carriage is established for conjugate polysaccharide vaccines but not for the MenB-targeting protein vaccines [212]. Since meningococci are able to downregulate capsule expression or switch capsules entirely [92], vaccine escape might be expected to occur. Similarly, high levels of variation are observed in the major components of both the Bexsero^®^ (fHbp, NadA, NHBA, and PorA [213]) and Trumenba^®^ (fHbp [214]) vaccines within the meningococcal population [67,215,216,217,218,219,220]. Variation in the glycan expressed on pilin and LOS structure has been proposed to occur in response to herd immunity in the meningococcal population [127,221,222], and it is clear from microevolution studies discussed previously that variation in antigenically visible structures is extremely common in outbreak settings. It is not unfeasible to expect a similar pattern of variation to occur in response to vaccination. In the wake of a broadly successful vaccination campaign against MenA in the African meningitis belt, an increase in the incidence of IMD caused by MenX and MenC occurred [223]. Similarly, MenB and MenY remain a major cause of disease in Europe and North America following successful reductions in the burden of MenC and MenW by vaccination [9]. A rise in MenB, MenE, and MenX carriage was recently reported in the Netherlands four years after the implementation of the MenACWY vaccine [48]. Whether these trends are reflective of selection for nonvaccine serogroups, the expansion of existing lineages into the niche left by the vaccine-controlled serogroups, or simply a part of the normal cycling of dominant meningococcal lineages is not clear. Further monitoring of meningococcal epidemiology as MenB vaccines become more widely used and a pentavalent MenACWYX vaccine is introduced in Africa will shed more light on the future of vaccination against meningococcal disease [224].

Capsule switching has been raised as a possible mechanism for the emergence of hypervirulent vaccine-escape variants [164]. However, capsule-switched isolates tend to cause localised disease clusters rather than outbreaks on a global scale. The key exception to this trend is the global success of MenW:cc11. Two studies compared the virulence of capsule-switched variants. A study comparing MenA, MenB, MenC, and MenX isolates from ST-7 in China found that MenA isolates were the most inflammatory and pathogenic in lung epithelial cells [225]. Since ST-7 isolates are most commonly MenA, this may imply that this lineage is adapted to the presence of the MenA capsule and that expression of alternative capsule polysaccharides may impose a fitness cost even if they provide a short-term benefit in terms of immune evasion. Another study compared the pathogenicity of isogenic capsule-switched mutants of strain H44/76 (MenB:cc32) expressing MenC, MenW, or MenY capsules [226]. This study also found that infection with H44/76 expressing a wild type MenB capsule caused greater neutrophil depletion and decreased survival in a zebrafish model of infection. The authors demonstrated a negative relationship between the pathogenicity of each mutant and the number of carbon atoms contained in the repeating unit of the associated capsule type [226]. It may therefore be expected that expression of different capsules places a different metabolic burden on the bacterial cell. Since the meningococcal capsule is also known to interfere with attachment and invasion of meningococci into host cells (especially those mediated by Opa proteins) [92,114,227], it seems likely that capsule-switched strains normally have a fitness defect in the absence of compensatory mutations which could restore hypervirulence.

## 6. Conclusions and Perspectives

Our understanding of meningococcal virulence and evolution is hampered by the sheer diversity generated by meningococcal genetics. Studies of meningococcal microevolution are key to teasing out the differences in niche adaptation between meningococcal lineages and how each lineage adapts to the pressures placed on it by the host environment. A somewhat disturbing trend among studies of microevolution is the repeated tendency of microevolution to generate new invasive variants, some of which may have the potential to escape vaccine-induced immunity. In particular, the increasing use of conjugate polysaccharide vaccines will place a significant selection pressure on the meningococcal population, which has the potential to drive evolution towards increased transmission fitness and capsule switching. This has the potential to generate hypervirulent meningococcal lineages for which our current vaccines are ineffective. Although the selective pressure imparted by the multivalent protein vaccines like Bexsero^®^ and Trumenba^®^ is significantly less since they do not prevent carriage, the circulation of hypervirulent isolates covered by these vaccines will continue. This in turn may serve as a genetic reservoir for the generation of new hypervirulent lineages which are not covered by these vaccines. The recent evolution of a capsule-null meningococcal lineage now adapted to a different ecological niche in humans, that is, in the urogenital tract, illustrates the flexibility of this pathogen to remove the expression of the capsule entirely but still retain its ability to transmit and cause disease in the human population. Continued genomic monitoring of the meningococcal population, along with more detailed studies of microevolution, is essential in order to identify emerging meningococcal lineages before they present a serious threat to public health.

## Figures and Tables

**Figure 1 microorganisms-11-03005-f001:**
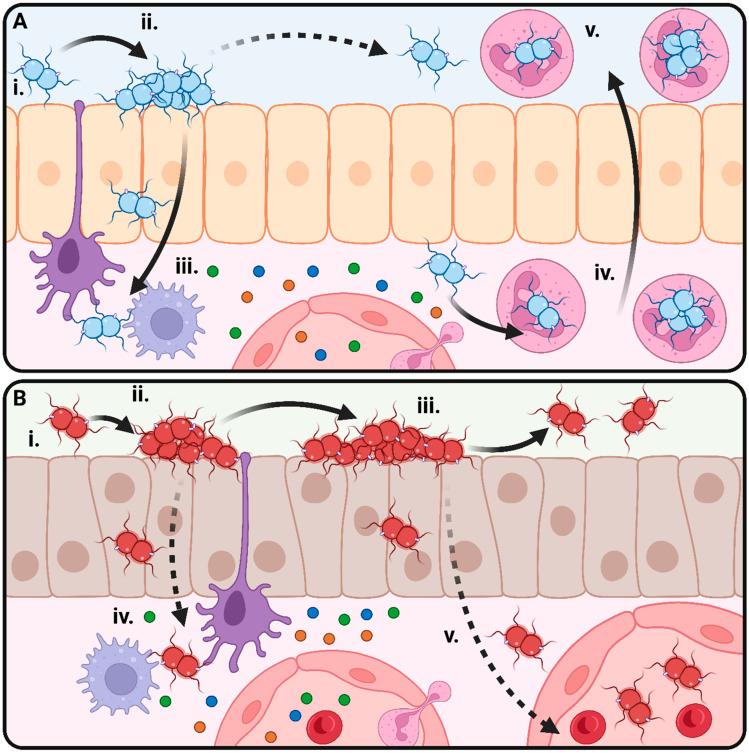
Lifestyles of the pathogenic *Neisseria*. (**A**) Lifestyle of *N. gonorrhoeae* (Ngo). (i) Initial attachment between Ngo and the urogenital epithelium is mediated by the type IV pilus. (ii) More intimate attachment is mediated by the adhesins including the opacity proteins and results in microcolony formation. (iii) Ngo crosses the epithelial barrier by transcytosis. Immune activation of resident macrophages and dendritic cells results in the release of cytokines. (iv) Neutrophils are attracted to the site of infection by cytokines and chemokines. Ngo initiates “silent” uptake into neutrophils and establishes a replicative niche inside neutrophils. (v) Infected neutrophils migrate to the epithelial surface. A neutrophil-rich purulent exudate facilitates further transmission to other hosts. (**B**) Lifestyle of *N. meningitidis* (Nme). (i) Initial attachment between Ngo and the urogenital epithelium is mediated by the type IV pilus. (ii) More intimate attachment is mediated by the adhesins including the opacity proteins and results in microcolony formation. (iii) Microcolonies mature, and dispersal of Nme results in transmission to other hosts through contact with aerosolised droplets. (iv) Occasionally, Nme will cross the epithelial barrier by transcytosis, stimulating a local immune response. (v) If a given strain of Nme can resist killing by the host immune system, it may enter the systemic circulation and cause invasive meningococcal disease.

**Figure 2 microorganisms-11-03005-f002:**
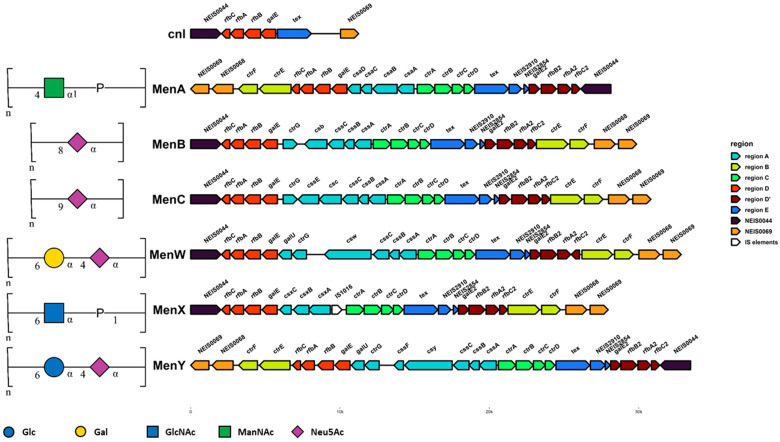
Capsule polysaccharide structure and genetic organisation of the *cps* island in the six clinically relevant meningococcal serogroups. The genetic organisation of the *cps* islands in representative closed genomes from MenA (Z2491), MenB (H44/76), MenC (FAM18), MenW (EXNM741), MenX (KL11168), and MenY (M23580) are shown along with the organisation of the *cnl* locus (α14). Genes are coloured by genetic region. Region A encodes polysaccharide synthesis, region B encodes capsule translocation, region C encodes capsule transport, region D encodes genes for LOS synthesis, region D′ is a degenerate copy of region D, and region E encodes the *tex* locus and two methyltransferase pseudogenes. Polysaccharide structures are drawn using the IUPAC Symbol Nomenclature for Glycans. Glc = glucose; Gal = galactose; GlcNAc = *N*-acetylglucosamine; ManNAc = *N*-acetylmannosamine; Neu5Ac = *N*-acetylneuraminic acid, sialic acid.

**Figure 3 microorganisms-11-03005-f003:**
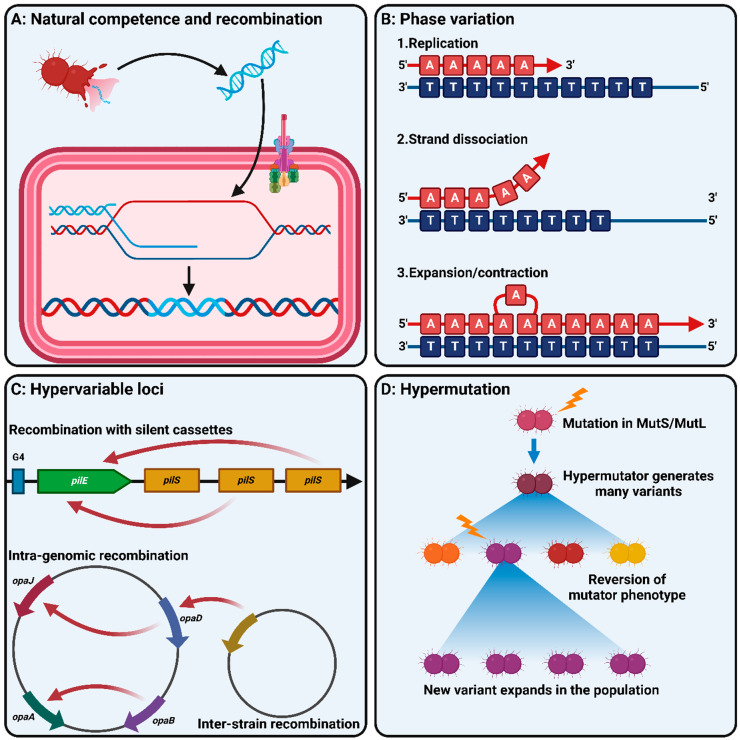
Mechanisms producing genetic variation in *N. meningitidis*. (**A**) Natural competence allows uptake of related DNA through the type IV pilus machinery. This DNA can undergo recombination with the chromosome to aid in DNA repair or to introduce novel genetic variants. (**B**) Phase variation in *Neisseria* occurs primarily by slipped-stranded mispairing of repetitive DNA elements during replication. As the repeat element is replicated, the newly synthesised strand may dissociate from the template strand and reanneal incorrectly, resulting in either contraction or expansion of the repeat element. Such changes in open reading frames result in the phasing of gene expression on or off completely, whereas changes in promoter regions may modulate the level of expression for the gene. (**C**) Hypervariable loci are genes which undergo frequent recombination, either with an array of silent pseudogenes (as is the case for the pilin gene *pilE*) or by recombination between homologous loci either within the same genome or on DNA acquired by natural transformation from other *Neisseria* spp. or closely related strains (as is the case for the *opa* loci). (**D**) Loss-of-function mutation of the DNA mismatch repair system MutSL results in a hypermutator phenotype which rapidly generates many novel variants. Eventually, a variant with an adaptive advantage may undergo reversion of the mutator phenotype and continue to expand within the population.

**Figure 4 microorganisms-11-03005-f004:**
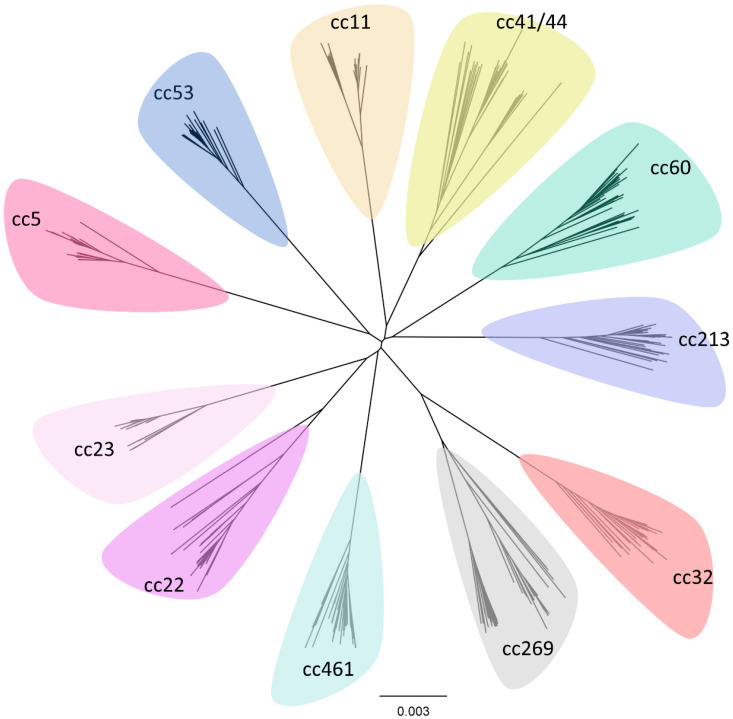
Clonal structure of the meningococcal population. A neighbour-joining phylogeny of 550 *N. meningitidis* isolates from 11 clonal complexes based on the phylogeny of 1735 core genes. Clonal complexes are each indicated by a different colour. Used with permission from Mullally, Mikucki, Wise, and Kahler [40].

**Figure 5 microorganisms-11-03005-f005:**
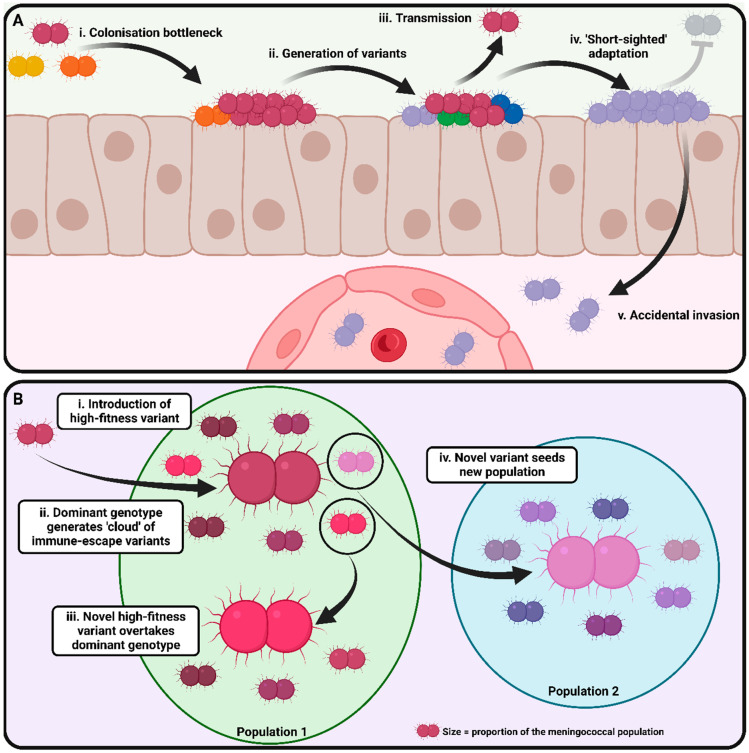
Microevolution in *N. meningitidis*. (**A**) Within-host evolution. (i) Colonisation of a new host presents a genetic bottleneck through which only some meningococcal variants will survive. (ii) Once colonised, microevolution events result in the production of multiple variants within the host. (iii) Some variants are fit in the wider population and will transmit to other hosts. (iv) Some variants have a survival advantage within a given host despite being less fit for transmission. (v) Occasionally, host-adapted variants will be coincidentally adapted for survival in the systemic circulation and go on to cause disease. (**B**) Microevolution within outbreaks/epidemics. (i) A high-fitness meningococcal variant enters a host population and becomes established as the dominant genotype. (ii) Microevolution events produce a ‘cloud’ of variants with a transient survival advantage in a small number of hosts due to immune-escape or colonisation bottlenecks. These variants are usually less fit and do not cause outbreaks of their own. (iii) Occasionally, a novel variant will have a broader survival advantage and overtake the previous dominant variant. (iv) Novel variants may be transmitted to new host populations, where they will become dominant through the founder effect or because they are adapted to the new host population.

**Table 1 microorganisms-11-03005-t001:** Cases of putative vaccine escape in bacterial pathogens of humans and animals.

Species	Vaccine(s)	Does Vaccination Protect against Colonisation and Transmission?	Proposed Mechanism of Vaccine Escape	Impact of Vaccination on Disease Rates	Notes	Reference
*Neisseria* *meningitidis*	Conjugate polysaccharide, OMVs	Conjugate vaccines most likely prevent or reduce carriage, but OMV vaccines do not.	Capsule switching, antigenic variation.	Global disease incidence has decreased, changes in dominant serogroups and lineages.		See text
*Haemophilus* *influenzae* serotype b	Conjugate polysaccharide	No.	Loss of capsule expression, serotype replacement, possible increased transmission/colonisation of Hib.	Recent increases in the Netherlands.		[191,192,193,194,205,206]
*Streptococcus pneumoniae*	Conjugate polysaccharide vaccines	Incomplete control of carriage.	Serotype replacement, loss of capsule expression, capsule switching.	Increase in prevalence of nonvaccine serotypes.		[187,188,189,190,206,207,208]
*Bordetella* *pertussis*	Killed whole-cell vaccine	No.	Loss of pertactin antigen.	Recent increases in global disease rates due to pertactin-deficient strains. A decrease in fitness of pertactin-deficient strains is observed.	A live-attenuated pertactin-deficient vaccine is already in development.	[196,197,198]
*Corynebacterium diphtheriae*	Diphtheria toxoid vaccine	No.	Antigenic variation of *tox*.	Recent global increases, unclear if vaccine escape is playing a role.		[195,199,209]
*Yersinia pestis*	Killed whole-cell vaccine		Loss of F1 pili expression due to IS element insertion.		Vaccine largely discontinued due to short-lived protection. Concerns about F1 expression largely impact host natural immunity.	[210,211]
*Pasturella* *multocida*	Killed whole-cell vaccine	-	Phase variation of LPS structures.		Causative agent of fowl cholera, vaccine used in poultry farming.	[204]
*Streptococcus* *iniae*	Killed whole-cell vaccine	-	Capsule antigenic variation.		Pathogen of farmed fish.	[202,203]
*Yersinia ruckeri*	Killed whole-cell vaccine	-	Loss of flagella expression.		Pathogen of farmed fish.	[200,201]

## Data Availability

No new data were created or analysed in this study. Data sharing is not applicable to this article.

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
