# Peer review of "Microevolution and Its Impact on Hypervirulence, Antimicrobial Resistance, and Vaccine Escape in Neisseria meningitidis"

_microorganisms, 2023, doi:10.3390/microorganisms11123005_

Round 1

Reviewer 1 Report

Comments and Suggestions for Authors

The authors proposed a review article that explores the genetic diversity, microevolution, and hypervirulence in Neisseria meningitidis, a bacterium that can cause meningitis and sepsis. The article discussed the mechanisms that allow the meningococcus to adapt and evolve, including horizontal gene transfer, recombination, and mutation. It also discussed the role of microevolution in shaping the evolutionary history and population structure of the meningococcus, and the potential for microevolution which can contribute to the escape of vaccines.  the article provided a valuable comprehensive overview of the ways in which the meningococcus is able to evolve and adapt, making it a highly diverse and adaptable organism.

It is a good review.   Here are only some suggestions on  some minor points of the article.

1. The mechanism of the generation of penicillin resistance in Ngo and Nme strains needs more explanation (Paragraph 3.1).

2. Could the authors provide a summary of the roles of the repeat families in recombination to the reader who are not familiar with the genetics of bacteria? (Paragraph 3.1)

3. The mechanism for SSRs to influence the phase variation needs more references beside ref.84  (Paragraph 3.2).

Author Response

 We thank reviewer 1 for their positive feedback and summary, our response to their remarks is outlined below:

  1. The mechanism of the generation of penicillin resistance in Ngo and Nme strains needs more explanation (Paragraph 3.1).

 In combination with the comments from reviewer 2 we have moved the information about penicillin resistance from this section into a section labelled “Microevolution in response to antimicrobial therapy” which describes the penicillin resistance mechanism more explicitly along with some commentary about microevolution resulting from antimicrobial usage (line 816-867).

  1. Could the authors provide a summary of the roles of the repeat families in recombination to the reader who are not familiar with the genetics of bacteria? (Paragraph 3.1)

 We have added a sentence explaining the role of repeat regions in providing the necessary homology for recombination (line 295): “In addition to DUS copies, the Neisseria genome contains a number of other repeat families which form regions around the genome with high levels of similarity to each other. Since recombination is highly dependent on homology, these regions form recombination hotspots and in some cases mediate genetic mobilization.”

  1. The mechanism for SSRs to influence the phase variation needs more references beside ref.84  (Paragraph 3.2).

We have added two references by Levinson et al. and Sinden et al. (Refs 85 and 86, line 345).

Reviewer 2 Report

Comments and Suggestions for Authors

Mikucki and Kahler:

The authors present an extensive review about genetic diversity, microevolution and hypervirulence in Neisseria meningitidis. They talk deeply about the pathogenicity of this species, genetic variation, epidemiology and host invasion. They describe various known mechanisms of immune evasion such as capsule switching, recombination, phase variation, etc. They also talk about microevolution within host, during vaccination as well as during epidemics. 

Overall, I found the review very extensive, well-written and thorough. I recommend publication. The only point I would add, if space is available, is to connect any current in-clinic treatment strategies of Neisseria species with reference to proteins or pathogen systems that are targeted by such clinical drugs.

Author Response

We thank reviewer 2 for their kind remarks. To address the clinical angle, we have added the section “Microevolution in response to antimicrobial therapy” to the review (line 816-867). This section is short since this review is focussed on the meningococcus in which AMR is rare, and many mechanisms of AMR in the related N. gonorrhoeae fall outside of the realm of microevolution.

Reviewer 3 Report

Comments and Suggestions for Authors

The sections "The pathogenic Neisseria and their relationship to the human host" and "Microevolution in Nme" need figures that visualize the mechanisms they describe.

Author Response

This is a good suggestion, we have added three new figures (now named figure 1, figure 3, and figure 5) to visualise some of the concepts in the review.